# CWPS: Efficient Channel-Wise Parameter Sharing for Knowledge Transfer

## Abstract

Knowledge transfer aims to apply existing knowledge to different tasks or new data, and it has extensive applications in multi-domain and multi-task learning. The key to this task is quickly identifying a fine-grained object for knowledge sharing and efficiently transferring knowledge. Current methods, such as fine-tuning, layer-wise parameter sharing, and task-specific adapters, only offer coarse-grained sharing solutions and struggle to effectively search for shared parameters, thus hindering the performance and efficiency of knowledge transfer. To address these issues, we propose Channel-Wise Parameter Sharing (CWPS), a novel fine-grained parameter-sharing method for Knowledge Transfer, which is efficient for parameter sharing, comprehensive, and plug-and-play. For the coarse-grained problem, we first achieve fine-grained parameter sharing by refining the granularity of shared parameters from the level of layers to the level of neurons. The knowledge learned from previous tasks can be utilized through the explicit composition of the model neurons. Besides, we promote an effective search strategy to minimize computational costs, simplifying the process of determining shared weights. In addition, our CWPS has strong composability and generalization ability, which theoretically can be applied to any network consisting of linear and convolution layers. We introduce several datasets in both incremental learning and multi-task learning scenarios. Our method has achieved state-of-the-art precision-to-parameter ratio performance with various backbones, demonstrating its efficiency and versatility.

## 1 Introduction

Knowledge transfer is a crucial task that involves extracting valuable information from existing tasks to enhance performance in new tasks or data. Existing knowledge can be repurposed and applied to new problems, offering a wide range of potential applications. For example, adapting a supernet to a new domain can be efficiently achieved through knowledge transfer (Zhuang et al., 2020). Two common and correlated task settings among these applications are Multi-Domain Learning (MDL) and Multi-Task Learning (MTL). MDL transfers knowledge from one domain to the target domain, while MTL transfers knowledge between several target tasks. Both involve knowledge transfer during training, which requires an efficient method.

However, it is not easy to design an algorithm that creates an appropriate model, shares knowledge between tasks without introducing too many parameters, and has high universality (Gesmundo & Dean, 2022). The algorithm needs to identify the level of granularity of parameter sharing, measure the relations of tasks, and operate at the unified level of neural networks, which can be challenging to balance. In addition, the consumption of extra training for knowledge transfer is also of concern, which puts more stringent demands on our algorithm. Therefore, we deduce that the key point is identifying a suitable shared object and efficiently measuring the relations between various tasks.

The existing coarse and parameter-efficient method fixes the parameters of the base model and then adds some trainable task-specific output layers (Vandenhende et al., 2020) or adapter layers (Rebuffi et al., 2017), sharing the entire backbone. Some other works explore ways to optimize parameter usage by investigating finer layer-wise parameter sharing (Gesmundo & Dean, 2022; Wallingford et al., 2022; Lu et al., 2017) and finest single-weight-wise parameter sharing (Mallya et al., 2018). However, neurons are the smallest computational units in neural networks and are also considered the smallest memory units of knowledge. While layer-wise parameter sharing provides a coarse-grained

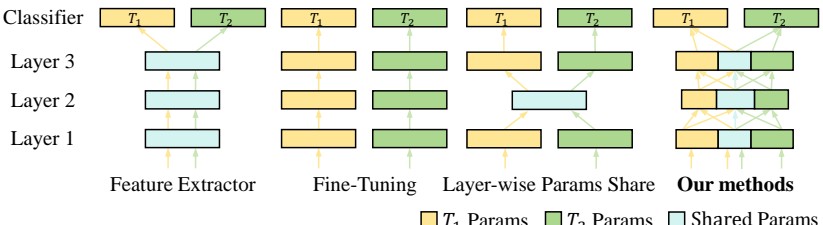

**Figure 1:** *The difference between different parameter sharing methods.* Two candidate tasks, $T_1$ and $T_2$, have task-specific parameters represented by green and yellow triangles respectively, while blue triangles represent shared parameters. Our method shares channels between layers, while others share layers between models.

knowledge transfer approach, they lack flexibility and cannot perform fine-grained knowledge sharing at the neuron level. Finest-grained methods split the weight parameters of each neuron in the network to share knowledge (Mallya et al., 2018). Still, neglecting the "atomic" nature of the neuron as the smallest memory unit has led to a huge parameter search overhead and suppressed knowledge transfer performance.

Further, heuristic algorithms (Gesmundo & Dean, 2022) are used to search for parent models to solve the enormous search space of shared objects(layers or weights), while Wallingford et al. (2022); Mallya et al. (2018) simplify the search process using learnable parameters. However, these methods to model the relations between tasks are costly and arbitrarily assume the most related base model, leading to inefficient and weak knowledge exchange between tasks.

To conclude, the decision to select the right level of granularity and solve task relations' search space is a dilemma: a coarser level of granularity reduces the search space but is less efficient, and a finer level of granularity provides more precise control but increases the search space. So, a natural question arises: can we seek a solution that efficiently determines the fine-grained sharing of objects?

In this paper, we propose a Channel-Wise Parameter Sharing (CWPS) method that avoids the dilemma of granularity and search space. In theory, our method is defined by the finest parameter-sharing unit, which is a single neuron. The neuron, also known as the channel, serves as the minimal concept of the neural network and has not been carefully explored by other parameter-sharing methods. As shown in Figure 1, incorporating channel-wise parameter-sharing mechanisms allows fine-grained control of task-specific parameters without compromising neuron consistency, resulting in more efficient parameter sharing. Besides, we propose a searching method called Composite Parent Model Searching(CPMS), which constructs a composite parent model for every search process by measuring parent and child weights. This search method greatly reduces the search space fine-grained control brought and naturally measures the relations between tasks. Further, CWPS operates at the fundamental network level, making it applicable in a wide range of scenarios. As long as the backbone comprises linear and convolution layers, CWPS can be utilized. Its universal nature enables a seamless transition from various task settings through iterations.

We measure our method in ImageNet-to-Sketch incremental learning benchmark (Berriel et al., 2019) and DomainNet multi-task learning benchmark (Peng et al., 2019). CWPS can transfer from one domain to another and reach state-of-the-art performance in incremental and multi-task learning scenarios. The contributions of our work are as follows:

1. We introduce CWPS, a fine-grained method for knowledge transfer across tasks. Compared to other parameter-sharing methods, it offers a more natural perspective and allows for the optimal and efficient utilization of neurons across various tasks.

2. We have developed a rapid architecture search algorithm called CPMS to determine shared parameters. This algorithm simultaneously models the relationships between different tasks.

3. CWPS is highly scalable, allowing simultaneous use in various knowledge transfer tasks to achieve state-of-art performance, and is adaptable to multiple network structures.

## 2 RELATED WORK

### 2.1 MULTI-TASK LEARNING

Multi-task Learning (MTL) is a transfer learning strategy that aims to enhance overall performance by concurrently addressing and learning from multiple correlated tasks within the same model. The main classification of multi-task frameworks is based on whether they are encoder-based or decoder-based. As exemplified by Cross-stitch Networks (Misra et al., 2016), encoder-based designs Misra et al. (2016); Liu et al. (2019); Gao et al. (2019); Lu et al. (2017) involve the sharing of task-specific features during the encoding phase. Subsequently, each task possesses an independent task head for processing the shared encoder features during decoding. Conversely, decoder-based multi-task frameworks Vandenhende et al. (2020); Xu et al. (2018); Zhang et al. (2018); Neseem et al. (2023) address all tasks directly within the same stage, acquiring outputs from all tasks in parallel or sequentially. Moreover, studies (Bhattacharjee et al., 2022; Sun et al., 2020) demonstrate the sharing of feature information in both the encoding and decoding stages simultaneously. These three frameworks are efficient in their respective task settings, but their specificity limits their universal applicability.

### 2.2 MULTI-DOMAIN LEARNING AND INCREMENTAL LEARNING

Multi-domain learning focuses on overcoming data disparities between domains to enhance model adaptability. It aims to address knowledge transfer between domains, resolving the challenge of using multiple datasets with differing statistical properties for the same task. Rebuffi et al. (2017) proposed the residual adapter method, incorporating the adapter into data representation from multiple domains to identify shared features in less similar domains. Subsequent algorithms Rebuffi et al. (2018); Rosenfeld & Tsotsos (2018), building upon the adapter concept, further refined it to achieve parameter efficiency while maintaining performance in the original domain. Other methods (Mallya et al., 2018; Mancini et al., 2018) utilize binary masks for simpler and more efficient MDL, but they balance reducing the parameter count and performance. Task Adaptive Parameter Sharing (TAPS)(Wallingford et al., 2022) tries to select the minimal subset of existing layers and retrain them. However, the layer-level parameter sharing still falls short of the idealized small parameter count.

Meanwhile, Multi-Domain Learning aims to transfer knowledge without forgetting the previous knowledge, which remains the same as common incremental learning. Consequently, the primary challenge MDL and incremental learning must address is mitigating catastrophic forgetting. Various methods have been proposed to address such issues, including functional and parameter regularization (Aljundi et al., 2018; Kirkpatrick et al., 2017; Schwarz et al., 2018), developing incrementally updated components (Li et al., 2024; Zhang et al., 2020), improving representation learning with additional inductive biases (Cha et al., 2021; Ni et al., 2023; 2021), they are suggestive to MDL.

### 2.3 MULTI-DOMAIN LEARNING VS. MULTI-TASK LEARNING

To conclude, we could summarize that MDL is MTL in very restricted situations. Multi-task learning learns different tasks without strict task constraints, while MDL learns one by one without revisiting prior data. Our objective for MDL aligns with the approach described in Wallingford et al. (2022), which involves starting from a pre-trained base model and then learning new tasks or domains. On the other hand, the aim of MTL is similar to the method outlined in Gesmundo & Dean (2022), where the final models are obtained iteratively through multiple datasets.

## 3 METHODOLOGY

Based on the conclusion in Section 2.3, our method begins with the restricted situation (MDL) and is then extended to a broader range of scenarios to facilitate knowledge transfer. For an incremental task, we first use CPMS to efficiently search and calculate the relations between the knowledge contained in the model of the previous task and the new task, and determine the relations between the neurons of the new task and each neuron of the previous task. Then, we use our fine-grained parameter sharing method, CWPS, to obtain the knowledge of the previous task, and fine-tune the model for a few training epochs. We iterate this process for each new task, thus completing the MDL.

**Figure 2:** *The composition of a single convolution $layer_i$ during training.* The cuboids in color are kernels of the convolution layer. Each color represents one task. This portion displays the layer's training structure, composed of the child kernels in green, the composite parent kernels in other colors, and $mask_{t,i}$ in gray. The composite parent kernels are composed of kernels from $W_{trained}$, which are untrainable, while child kernels and $mask_{t,i}$ are trainable.

### 3.1 PROBLEM STATEMENT

Previous works have shown their effect on sharing weight with the pre-trained model. Following Gesmundo & Dean (2022), when training models for new tasks, the trained models are named the parent models while the child model is used for new tasks. Assuming that there are $K$ tasks:

$$\tau = \{T_1, T_2, ..., T_K\} \tag{1}$$

and one task $T_0$ of the pre-trained model, we need to train $K$ models for every task while introducing as few task-specific parameters as possible.

Before sharing parameters, some layers, such as Batchnorm2d, contain a small number of parameters (less than 1% in most networks) and may deteriorate performance if shared. Thus, we do not need to exchange their parameters among tasks. Besides, the output layers (the last linear layer for classification tasks) should be task-specific since the output shape differs from task to task.

### 3.2 EFFICIENT KNOWLEDGE TRANSFER FOR MULTI-DOMAIN LEARNING

#### 3.2.1 DESIGN MOTIVATIONS

To make our method universally applicable, we work at the basic level of the network, which includes the linear and convolution layers. As shown in Figure 2, our algorithm for the $layer_i$ from task $t$'s model involves trainable child parameters $w_{t,i}$ and $b_{t,i}$, untrainable parent parameters $w'_{t,i}$ and $b'_{t,i}$ from other tasks' trained models, and a $mask_{t,i}$ of the out-channels size to control parameter sharing. However, it can be challenging to determine the origin of these parent parameters. As a result, we propose a composite parent model, which allows us to gather information from all potential parent models and reduce the search space.

#### 3.2.2 COMPOSITE PARENT MODEL SEARCHING

The weight of all the trained layers (linear, conv) from existing tasks, apart from those task-specific layers mentioned above, is collected in the set $W_{trained}$. Given a new task $t \in \tau$, all parameters in the set are available to get the model for this task. After training the model for $t$, the corresponding weights will be added to the set. Consequently, for the first task $T_1$, $W_{trained}$ only contains the weights from the pre-trained model (such as the pre-trained model in ImageNet). Thus, it can be initialized by

$$W_{trained} = \{w_{T_0,1}, b_{T_0,1}, w_{T_0,2}, b_{T_0,2}, ..., w_{T_0,N}, b_{T_0,N}\} \tag{2}$$

where $w_{t,i}$ and $b_{t,i}$ are the parameters of the $i$'th layer for task $t$ and $N$ is the number of all the shareable layers for a model.

Before assigning the weight of the parent model, several preparations are needed. First, we utilize the typical finetune settings to train the new model for a few epochs (one-quarter of the total training epochs). This phase allows us to get rough references to get the parent weight, which can be completed at a trivial cost.

The searching process is shown in Figure 3. For every kernel $j \in \{1, 2, ..., C_i\}$ ($C_i$ is the number of output channels of $layer_i$) of $w_{t,i}$ ($t \in \tau, i \in \{1, 2, ..., N\}$), we obtain the task $\tilde{t}$ where the parent

**Figure 3:** *(a) The pipeline of our method.* The figure contains large rectangles representing layers, each composed of many small rectangles representing channels. A mask is applied to each channel during the training stage. The colors of the layers represent the parameters from their respective tasks. This figure illustrates the algorithmic execution process of the model for the task, which is represented by green. *(b) The composite parent model selection process for layer $i$ of task $T_k$.* The green kernels are reference child kernels (shown in the blue double-dashed box), while the others are parent kernels to be selected. Our method calculates the similarity between two kernels (shown in the blue lines) using the function $D$ in Eq. 3, and then identifies the most likely parent kernel from other tasks (shown in the blue dotted box). Afterward, we get the final updated kernels by $mask_{t,i}$. Kernels in $W_{trained}$ outlined with dashed lines indicate parameters sharing with previous tasks.

weight from follows:

$$\tilde{t} = \underset{t' \in \tau, t' \neq t, w_{t',i} \in W_{trained}}{\arg\min} D(w_{t',i}[j,:], w_{t,i}[j,:]) \tag{3}$$

where $D$ is the function to measure the similarity between the weight of two kernels (we use L2 distance and cosine similarity in practice). Thus, the weight of $j$'th channel is assigned by:

$$w'_{t,i}[j,:] = w_{\tilde{t},i}[j,:] \tag{4}$$

$$b'_{t,i}[j] = b_{\tilde{t},i}[j] \tag{5}$$

After calculating all the kernels of $w'_{t,i}$ and $b'_{t,i}$ from 1 to $C_i$ iteratively, every channel is assigned the trained weights of one channel from $W_{trained}$. These trained weights combine to form the composite parent model in Figure 2. As can be deduced from Eq. (2-5), if the task is $T_1$, all the parent weights are from the pre-trained model. For other tasks, the parent weights come from all the models trained for the prior task.

### 3.2.3 Weight Parametrization

Motivated by the mask generation method of Yan et al. (2021), we introduce a channel mask $mask_{t,i} \in R^{C_i}$ to control which channel to share. Assuming $layer_i$ is a linear layer, its weights are updated as follows:

$$w''_{t,i} = (mask_{t,i} \cdot w^T_{t,i} + (1 - mask_{t,i}) \cdot w'^T_{t,i})^T \tag{6}$$

$$b''_{t,i} = mask_{t,i} \cdot b_{t,i} + (1 - mask_{t,i}) \cdot b'_{t,i} \tag{7}$$

where $w''_{t,i}$ and $b''_{t,i}$ are the updated parameters for the final child model.

However, some kernels in $w''_{t,i}$ should be trainable while others from trained models are untrainable, this makes it hard to optimize $w''_{t,i}$. To enable the back-propagation of the torch, the original inference phrase of the linear layer $L(x) = w_{t,i}x + b_{t,i}$ is finally converted to:

$$L'(x) = mask_{t,i} \cdot (w_{t,i}x + b_{t,i}) + (1 - mask_{t,i}) \cdot (w'_{t,i}x + b'_{t,i}) \tag{8}$$

---

**Algorithm 1** Multi-task learning mode CWPS algorithm

---

**Input**: Pre-trained model $m_0$ with $N$ layers, trained weight $W_{trained}$ = $\{w_{T_0,1}, b_{T_0,1}, w_{T_0,2}, b_{T_0,2}, ..., w_{T_0,N}, b_{T_0,N}\}$
**Parameter**: $\tau = \{T_1, T_2, ..., T_K\}, Iterations$
**Output**: Trained models $M$

1: $c_1 = c_2 = ... = c_K = 0, m_1 = m_2 = ... = m_K = None, M = \{\}$
2: **for** $iter = 1$ to $Iterations$ **do**
3:    **for** $t = T_1$ to $T_K$ **do**
4:       $c = 0, m = m_0$
5:       Following Section 3.2, get the model $m$ with the validation accuracy $c$
6:       **if** $c > c_i$ **then**
7:          $c_i = c$
8:          **if** $m_i \neq None$ **then**
9:             Remove weights from $m_i$ in $W_{trained}$         ▷ Exclude useless parameters
10:         **end if**
11:          Add weights from $m$ to $W_{trained}$    ▷ Parameters that can be shared with other task
12:          $m_i = m$                       ▷ Better model $m$ for $T_i$
13:       **end if**
14:    **end for**
15: **end for**
16: **return** $M = \{m_1, m_2, ..., m_K\}$

---

where $w_{t,i}$ and $b_{t,i}$ from trained models are frozen, $w'_{t,i}$ and $b'_{t,i}$ are trainable.

### 3.2.4 TRAINING STRATEGIES

The training procedure is divided into three stages: soft mask training stage, hard mask training stage, and post-training stage.

**Soft mask training stage.** In this stage, $mask_{t,i}$ is trainable to optimize it. The elements of $mask_{t,i}$ should fall into the interval $[0, 1]$, so the mask is assigned by:

$$mask_{t,i} = Sigmoid(s_{t,i}) \tag{9}$$

where $s_{t,i}$ is the learnable parameter and can be initiated according to the standardized results of function $D$ in Eq. (3). After a few training rounds (one-fourth of the total training epochs), we can use the straight-through estimator $s_{t,i}$ to prune network weights, drawing inspiration from TAPS.

**Hard mask training stage.** We serve $s_{t,i}$ as the channel-wise task-specific estimator for pruning the weights that can be shared between tasks following

$$mask_{t,i} = F(Sigmoid(s_{t,i})) \tag{10}$$

$$F(x) = \left\{ \begin{array}{l} 0, x < \lambda \\ 1, x \geq \lambda \end{array} \right. \tag{11}$$

where $\lambda$ is the threshold to control the ratio of the parameters shared with the parent model. The weight of the corresponding channel is task-specific if one element in $mask_{t,i}$ is 1, and vice versa. Following the typical way of pruning, several training epochs are needed when pruning the weights that can be shared. We assign three-fourths of the training epochs to get the task-specific weights in practice.

**Post-training stage.** After finishing training the network, all weights could be simplified following Eq. (6, 7) to get faster inference performance and fewer parameters. This part does not degrade the model's performance in the hard mask training stage and does not require further training; it merely discards those redundant parameters in inference determined by the soft mask training stage, as shown in Figure 3 (a). Thus, CWPS would not reduce the inference speed compared to fine-tuning.

| | Param Count | Param Efficient | Flowers | WikiArt | Sketch | Cars | CUB | Mean |
|---|---|---|---|---|---|---|---|---|
| Fine-Tuning | 6× | ✗ | 95.73 | 78.02 | 81.83 | 91.89 | 83.61 | 86.22 |
| Feature Extractor | 1× | ✓ | 89.14 | 61.74 | 65.90 | 55.52 | 63.46 | 67.15 |
| Spot-tune (Guo et al., 2019) | 7× | ✗ | 96.34 | 75.77 | 80.20 | 92.40 | 84.03 | 85.75 |
| Piggyback (Mallya et al., 2018) | 6× | ✗ | 94.76 | 71.33 | 79.91 | 89.62 | 81.59 | 83.44 |
| WTPB (Mancini et al., 2018) | 6× | ✗ | 96.50 | 74.80 | 80.20 | 91.50 | 82.60 | 85.12 |
| TAPS (Wallingford et al., 2022) | 4.12× | ✓ | **96.68** | **76.94** | 80.74 | 89.76 | 82.65 | 85.35 |
| BA$^2$ (Berriel et al., 2019) | 3.8× | ✓ | 95.74 | 72.32 | 79.28 | **92.14** | 81.19 | 84.13 |
| Packnet⟶ (Mallya & Lazebnik, 2018) | 1.6× | ✓ | 93.00 | 69.40 | 76.20 | 86.10 | 80.40 | 81.02 |
| Packnet⟵ (Mallya & Lazebnik, 2018) | 1.6× | ✓ | 90.60 | 70.30 | 78.70 | 80.00 | 71.40 | 78.20 |
| CWPS | 3.2× | ✓ | 94.74 | 76.25 | **81.29** | 91.80 | **83.90** | **85.60** |

Table 1: **The results of different incremental methods in ImageNet-to-Sketch benchmark using ResNet-50**. The *Param Count* column means the proportion of all model parameters to the parameters of the pre-trained backbone, and other digits represent the Top-1 accuracy. The results are categorized into three parts: The first part shows the lower and upper boundaries, representing typical fine-tuning methods and fixing the backbone. The methods of the second part introduce parameters no fewer than fine-tuning, and results with under-line are the most optimal task results among them. The last part presents the results of parameter-efficient methods, with the best results highlighted in bold.

### 3.3 ITERATIVE JOINT LEARNING

The difference between incremental learning and common multi-task learning is that the tasks come incrementally, and all data is available at once in incremental learning, while multi-task learning is not. So, to fully use the training data, many methods (Gesmundo & Dean, 2022; Lin et al., 2019) train the models iteratively, which means more opportunities to search for child model structures.

CWPS can be easily extended to any multi-task learning method from Multi-Domain learning for its flexibility. As shown in Algorithm 1, when changing CWPS into multi-task learning mode, the training iteration won't terminate after finishing training the model of $T_K$. Instead, $T_1$ is the next task to be dealt with, and we will then go through $\tau$ one or more times. During the traversal, we only save the model with the best performance (highest validation accuracy) for one task, and we replace the weights from the worst model in $W_{trained}$ with those from the better model. Thus, every iteration of the traversal will gain better models and the number of weights in $W_{trained}$ will be kept within specified bounds.

## 4 EXPERIMENTS

In this section, we will compare CWPS with existing methods. We adjust the pre-trained models separately for each task and integrate them to form a comprehensive model applicable to various domains. Our solution is validated on the ImageNet-to-Sketch benchmark (Berriel et al., 2019) and DomainNet benchmark (Peng et al., 2019). Due to space limitations, more details and discussions can be found in the Appendix A.

### 4.1 DATASETS AND METRICS

**ImageNet-to-Sketch benchmark.** For the ImageNet-to-Sketch benchmark, we adopted the evaluation methods from prior work (Guo et al., 2019; Mallya et al., 2018; Mallya & Lazebnik, 2018; Mancini et al., 2018; Wallingford et al., 2022). Starting with models pre-trained on the ImageNet (Deng et al., 2009) dataset (Russakovsky et al., 2015), we transferred them to five additional joint datasets for evaluation. These datasets include:, VGG-Flowers (Nilsback & Zisserman, 2008), Stanford Cars (Krause et al., 2013), Caltech-UCSD Birds (CUBS)(Welinder et al., 2010), Sketches (Eitz et al., 2012), and WikiArt (Saleh & Elgammal, 2015). These datasets vary significantly, encompassing various categories (e.g., cars, birds) and diverse image appearances (natural images, artistic paintings, sketches). We resized all images to 224 and applied random horizontal flipping as data augmentation during training.

| | Param Count | Flowers | WikiArt | Sketch | Cars | CUB | Mean |
|---|---|---|---|---|---|---|---|
| DenseNet-121 Fine-Tuning | 6× | 95.6 | 77.0 | 81.1 | 89.5 | 82.6 | 85.2 |
| Piggyback | 6× | 94.7 | 70.4 | 79.7 | 89.1 | 80.5 | 82.9 |
| TAPS | 3.7× | **95.8** | 73.6 | 80.2 | 88.0 | 80.9 | 83.7 |
| CWPS-Incremental | 3.1× | 93.9 | 77.6 | 80.2 | 89.5 | 81.1 | 84.46 |
| CWPS-Joint | 3.1× | 94.0 | **78.0** | **80.4** | **90.0** | **81.7** | **84.82** |

| Method | Real | Painting | Quickdraw | Clipart | Infograph | Sketch | Mean |
|---|---|---|---|---|---|---|---|
| TAPS | 76.47 | 65.21 | 52.87 | 76.15 | 35.07 | 66.54 | 62.05 |
| AdaShare | 78.71 | 64.01 | 67.00 | 73.07 | 31.19 | 63.40 | 62.90 |
| CWPS | **81.69** | **67.73** | **70.28** | **77.21** | **36.88** | **67.06** | **66.81** |

Table 2: **The results of multi-task training using ResNet-50 in ImageNet-to-Sketch benchmark** *(top)* **and DomainNet benchmark** *(bottom)***.** Similar to Table 1, these two tables simultaneously display the number of parameters and precision. The highest level of accuracy is indicated by bold text in their respective settings.

**DomainNet benchmark.** Domain Datasets (Peng et al., 2019) is a large-scale dataset designed for evaluating domain adaptation methods. It comprises approximately 600k images, spanning 345 categories across 6 distinct domains: Clipart, Infograph, Painting, Quickdraw, Real, and Sketch. This dataset encompasses various object categories such as furniture, clothing, electronics, mammals, and buildings, making it notable for its diversity. Compared to smaller, environment-specific datasets, it offers a more comprehensive benchmark for assessing domain adaptation models. Following the configuration and augmentation methods outlined in TAPS (Wallingford et al., 2022), we will utilize this dataset as a benchmark, treating each domain as an individual task and employing their official training and testing methodologies.

**Metrics.** To measure efficient methods, the metrics should be able to take into account the precision and the parameter count. Since our benchmarks consist of classification tasks, our methods are evaluated based on the average Top-1 accuracy for each task and the average accuracy across all tasks of several experiments. Moreover, the parameter count is calculated based on the proportion of all model parameters to the parameters of the pre-trained backbone.

## 4.2 MULTI-DOMAIN LEARNING

**Results on ImageNet-to-Sketch.** We first measure our method in the ImageNet-to-Sketch benchmark. As seen from Table 1 and Figure 4 (left), only two methods, Spot-tune and Fine-Tuning, get a better mean accuracy than CWPS. However, these two methods assign no less than 1× backbone parameters for a task, which means low parameter utilization. Other methods Wallingford et al. (2022); Berriel et al. (2019); Mallya & Lazebnik (2018), somehow sharing parameters between tasks, perform worse than CWPS in both mean accuracy and parameter efficiency. Although Packnet only uses 1.6× backbone parameters, its performance is unstable and relatively worse in these methods. In other words, CWPS makes full use of every convolution kernel in the backbone, thus striking a state-of-the-art balance between task accuracy and parameter count.

## 4.3 MULTI-TASK LEARNING

**The effect of multi-task training.** For fairness, the results in Section 4.2 are confined to incremental learning mode, which only allows us to go through all the datasets once. In this section, the effect of multi-task training is shown. We first compare our methods with previous incremental methods, and densenet-121 is chosen for the backbone. As seen in Table 2, the performance of all the tasks is dramatically improved when multi-task joint training. Besides, our method maintains parameter efficiency, and every task's average number of parameters remains the lowest.

**Results on DomainNet.** Apart from the multi-domain benchmark ImageNet-to-Sketch, we also measure CWPS on the DomainNet benchmark. As shown in the bottom table in Table 2, we compare our method with the multi-task learning method AdaShare Sun et al. (2020). CWPS shows efficiency in terms of both parameter and computational resources. In the case of ResNet-50, CWPS achieves greater accuracy while introducing fewer parameters, demonstrating a more comprehensive sharing of parameters.

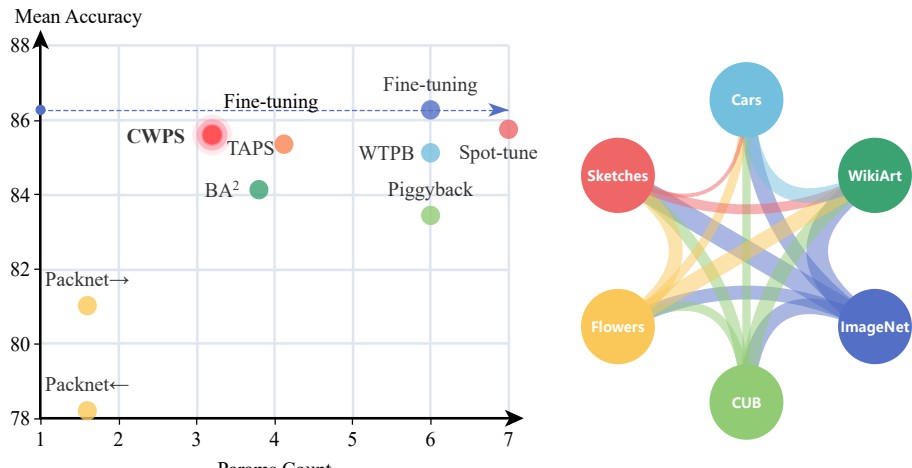

**Figure 4: The relationship between precision (Top-1 accuracy) and the number of parameters for different methods.** *(left)* Methods located in the top-left region generally exhibit superior performance. **The relation of different tasks in ImageNet-to-Sketch benchmark based on our algorithm.** *(right)* Each curve represents the relationship between the two tasks. Its thickness indicates the strength of the relationship. A higher number of shared neurons means a stronger relationship between tasks.

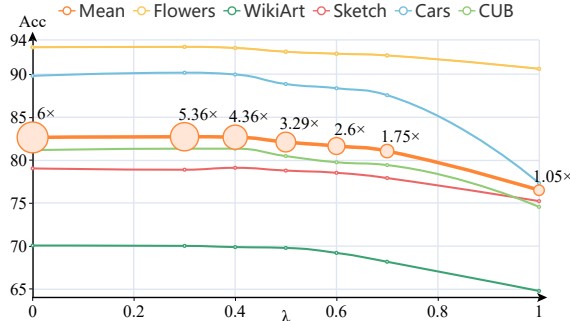

**Figure 5:** Diagram of the variation of model performance and model parameters with $\lambda$. This experiment used ResNet18 as the backbone, controlled the value of $\lambda$ from 1 to 0, and was trained for 3 iterations to obtain the results. Among them, the area of the orange empty circle represents the size of the model's parameters after completing the algorithm.

**Measure the relation between tasks**  In Figure 4 (right), the relation of tasks is quantified without direction. The thickness of the lines represents the degree of association between the data from the two datasets. For example, the Sketch dataset contains many categories such as "flower with stem", "leaf", and others that are directly semantically related to the Flowers dataset. In contrast, the Cars dataset lacks such categories related to the Flowers dataset. As a result, the connection between the Sketch and the Flowers should be stronger than between the Cars and the Flowers, which aligns with our experimental findings.

As shown in the Figure 4 (right), all tasks have stronger relations with ImageNet than any other tasks. This highlights the comprehensiveness of the ImageNet Dataset. Besides, the Sketches Dataset shows the weakest relations with other data domains. This is consistent with the ImageNet-to-Sketch benchmark design and our intuitive understanding of differences between data domains.

**The effect of $\lambda$**  We conduct the ablation experiment for $\lambda$ to further demonstrate the impact of our method in balancing the number of parameters and model performance. As shown in Table 3 and fig. 5, we can observe that when $\lambda$ is less than 0.3, the model's performance shows little improvement and even declines due to overfitting. On the other hand, when the value of $\lambda$ is greater than 0.7, the reduction in the number of parameters is very modest, but there is a significant drop in model performance.

| $\lambda$ | Param Count | Flowers | WikiArt | Sketch | Cars | CUB | Mean |
|-----|-------------|---------|---------|--------|------|-----|------|
| 0.0 | 6× | 93.14 | **70.05** | 79.03 | 89.81 | 81.16 | 82.64 |
| 0.3 | 5.36× | **93.19** | 70.01 | 78.88 | **90.18** | **81.33** | **82.72** |
| 0.4 | 4.36× | 93.06 | 69.87 | **79.10** | 89.97 | **81.33** | 82.67 |
| 0.5 | 3.29× | 92.62 | 69.77 | 78.78 | 88.86 | 80.47 | 82.10 |
| 0.6 | 2.60× | 92.39 | 69.19 | 78.52 | 88.37 | 79.75 | 81.64 |
| 0.7 | 1.75× | 92.19 | 68.16 | 77.91 | 87.56 | 79.42 | 81.05 |
| 1.0 | 1.05× | 90.63 | 64.76 | 75.22 | 77.31 | 74.55 | 76.49 |

**Table 3:** Our ablation studies of $\lambda$ use ResNet18 as the backbone, control the value of $\lambda$ from 1 to 0, and train for 3 iterations to obtain the results.

## 5 DISCUSSIONS

Although our approach focuses on parameter efficiency in incremental and multi-task learning, other methods, such as Mixture of Experts (MoE), model merging, pruning-based techniques, and prompt-based techniques, are intuitive. In this section, we will compare these methods with CWPS.

Some MoE-based methods, such as AdaMV-MoE (Chen et al., 2023) and M3ViT (Fan et al., 2022), address the issues of gradient interference in multi-task learning and the impact of task quantity on model inference speed to some extent. The MoE method involves simultaneous training of expert selection and multiple expert models, resulting in a dynamic model structure. However, while their inference structures are flexible, they often focus on a single transformer architecture and are not universally applicable across other structures compared to CWPS. Additionally, they only share parameters between layers and require multiple models to be pre-loaded during inference, leading to certain parameter waste.

Model merging (Matena & Raffel, 2022; Ilharco et al., 2022; Yadav et al., 2024) and our approach involve parameter sharing among multiple pre-trained models to adapt to different tasks. The method based on task vectors (Zhang et al., 2024) was very inspiring for our design approach. Our method essentially seeks to identify a finer-grained task vector that recognizes the underlying common information between tasks. Similar to MoE, the granularity and proportion of parameter sharing in model merging are relatively low, potentially leading to higher computational resource consumption during inference.

On the other hand, pruning-based methods, such as CPG (Hung et al., 2019) and BA$^2$ (Santos et al., 2022) subsequent approach, aim to reduce the number of parameters in multi-task settings, thereby improving inference speed. However, the lack of direct parameter sharing results in lower utilization of model parameters compared to parameter-sharing methods like ours, and the reduced computational load during inference can lead to decreased model performance.

Methods based on prompts (Wang et al., 2022) attempt to establish a prompt for each task and category to measure their correlations, quantifying the relationships between tasks. However, due to the limitations of the transformer architecture, this approach still has certain application constraints on other backbones, such as convolutional networks.

These methods aim to improve each task's performance by incorporating information from multiple domains, while our method prioritizes achieving a balance between parameter count and performance across multiple tasks.

## 6 CONCLUSION

This paper demonstrated Channel-Wise Parameter Sharing, a plug-and-play incremental and multi-task learning method. By fine-grained control and composite parent model, our method exhibits superior knowledge transfer capabilities, requiring fewer iterations and fewer parameters while maintaining high precision. Our experiments confirm these capabilities and demonstrate that CWPS can automatically adapt pre-trained models to various tasks and uncover relationships between tasks. Our methods have limitations, including increased video memory usage during training and requiring more steps to save models. Further research is needed.

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

# A APPENDIX

## A.1 MORE ABOUT EXPERIMENT

**Training details.** We use various ImageNet pre-trained models as the base models. For ResNet, we train the models for 40 epochs with batch size 96 in RTX4090, and the learning rate is from 0.01 to 0.04 in different circumstances. The learning rates for DenseNet (Iandola et al., 2014) and the vision transformer Dosovitskiy et al. (2020) are set to 0.08 and 0.001, respectively, while the other settings remain unchanged. One training iteration in the ImageNet-to-Sketch benchmark spends about four hours, while it spends about twenty-two hours in DomainNet benchmark. Throughout the training period, one-quarter of the epochs are dedicated to the soft mask training stage, while the remainder focuses on the hard mask training stage. During the soft mask training stage, the learning rate of mask parameter $s_i$ is set to 0.02. Besides, SGD optimizer and cosine learning rate scheduler are used to improve performance.

For CWPS hyperparameters, the threshold $\lambda$ is set from 0.3 to 0.7, so we can obtain a flexible control of the sharing ratio. In terms of the initialization of the mask parameter $s_i$, we normalize the values of $D(\cdot)$ in Eq. 3 and multiply them by $\frac{3}{mean(D(\cdot))}$, which helps us project these values on a reasonable interval.

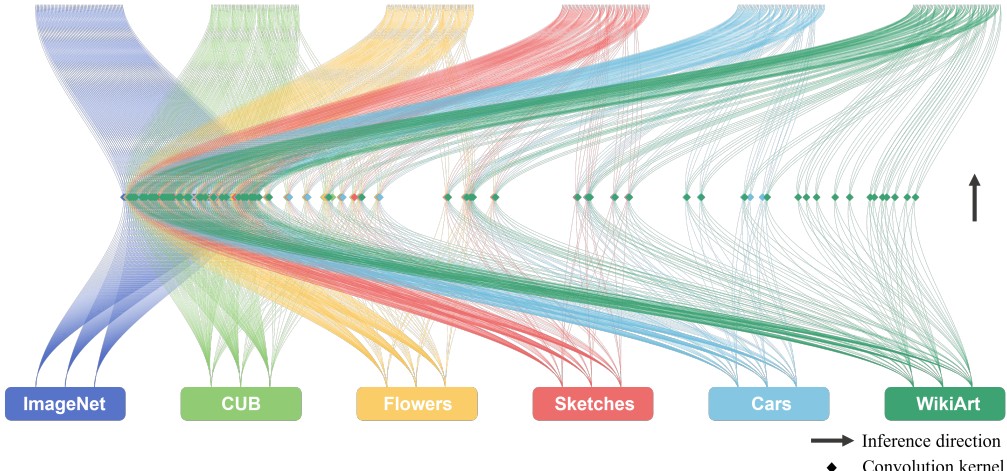

**Figure 6:** The kernels of the first layer of the supernet consisted of ResNet-18 (3 in-channels and 64 out-channels) for five tasks and ImageNet. The figure uses different colors to represent different tasks, diamonds to represent kernels, and lines to represent individual weights. If two kernels from different models share parameters, they are placed together.

**Task-specific parameters.** In Figure 6, we visualize the parameters sharing of the first layer in ImageNet-to-Sketch benchmark. After five training iterations, we get five ResNet-18 models for these benchmark tasks. Together with the ImageNet pre-trained model, the input layers of six models are visualized. As expected, kernels from different tasks are shared or are task-specific according to the preset threshold value $\lambda$. As these models are initialized using a pre-trained model from ImageNet, they will likely share parameters with the pre-trained model. Meanwhile, as the number of iterations grows, the number of actual task-specific parameters becomes fewer and fewer (only a dozen kernels are task-specific in this case) because the information on these tasks is more fully integrated. Finally, the illustration demonstrates how CWPS can reveal the connection between tasks kernel by kernel, allowing the child model to transfer knowledge from various datasets encountered effectively.

**The effect of iteration.** Figure 7 illustrates the progressive improvement in accuracy over the iterative process, reflecting the exchange of information between tasks. The second iteration demonstrates the most significant improvement in accuracy, attributed to the models in this iteration having access to information from all the tasks for the first time. From this fact, we deduced that these parent models in the second iteration are more comprehensively composed by parameters trained in all datasets, as intended by the design of the iterative mechanics. Besides, if the task settings are limited in computing resources, choosing two iterations would lead to the most economical method.

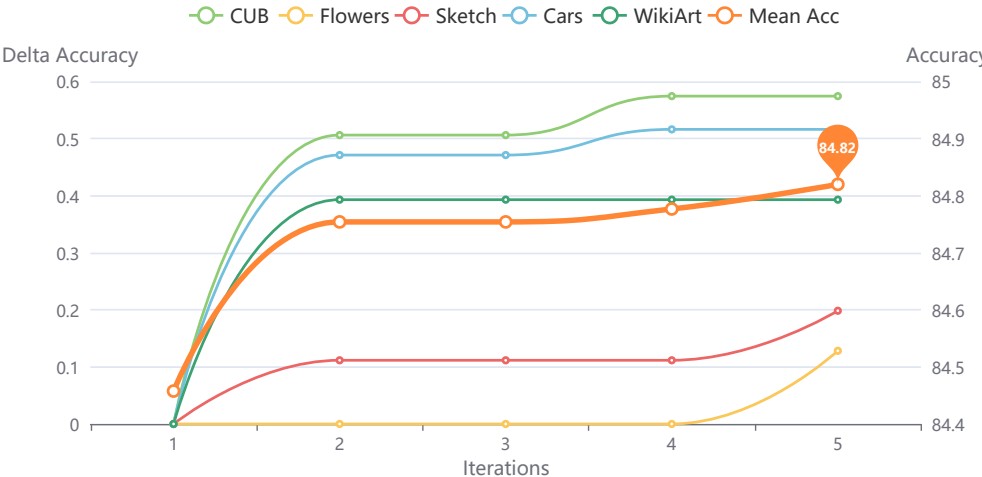

**Figure 7:** The figure illustrates the effect of iteration. The orange line represents the fluctuation of mean accuracy as the number of iterations increases, while the other lines depict the incremental Top-1 accuracy of each task from the first iteration. The final mean accuracy is pinned in the figure, which is obtained after five iterations.

|  | Flowers | WikiArt | Sketch | Cars | CUB |
|---|---|---|---|---|---|
| Random Init | 94.62 | 75.74 | 80.80 | 91.52 | 83.78 |
| Constant Init | - | - | - | - | - |
| Ours | 94.74 | 76.25 | 81.29 | 91.80 | 83.90 |

**Table 4:** The table displays the results of various mask initializations tested on the ImageNet-to-Sketch benchmark. The "Random Init" refers to initialization with a Gaussian distribution. The "Constant Init" refers to initialization with a constant value. Our method is mentioned in Appendix A.1

**The effect of mask initialization.** As depicted in Table 4, various mask initializations can lead to different outcomes, with Random Init performing poorly across all datasets. Meanwhile, if the mask is initialized with a constant value, its elements remain unchanged during training, resulting in an inappropriate initialization. This suggests that while using trainable parameters helps measure the potential for sharing each channel, a single parameter may not be sufficient to identify the optimal sharing structure within the entire search space. We experimented with various initialization methods (such as constant initialization, normal initialization, etc.) before ultimately selecting the approach outlined in the Appendix A.1. Therefore, it is essential to choose a suitable initialization method.

|  | Flowers | WikiArt | Sketch | Cars | CUB | mean |
|---|---|---|---|---|---|---|
| ViT Fine-Tuning | 99.3 | 82.6 | 81.9 | 89.2 | 88.9 | 88.4 |
| TAPS | 99.1 | 82.3 | 82.2 | 88.7 | 88.4 | 88.1 |
| CWPS | 99.0 | 81.2 | 82.4 | 89.4 | 88.7 | 88.1 |

**Table 5:** The results of ViT in ImageNet-to-Sketch Benchmark show that our method is competent with TAPS in mean accuracy. Meanwhile, TAPS's parameter count is about 4×, but our parameter count is 2.6×.

**The scalability of CWPS.** As shown in Table 5, CWPS can be easily applied to the transformer-based backbone without changing the structure. Our method utilized only 2.6× parameters to get the same performance as TAPS(its parameter count is not reported in detail). The basic structure of the transformer layer is the linear layer, which serves as the projection function to get q, k, and v. We deduce that the transformer (linear-based) layers are denser than the convolution layers, making it easier for our method to identify fine-grained sharing candidates. As a result, more parameters in transformer layers can be shared.

**The effect of the pre-trained models.** We assess the impact of the pre-trained models from two perspectives. The results will vary significantly if the base models are trained on different datasets,

for example, ResNet-50 trained on ImageNet and Places-365 Zhou et al. (2017) (the pre-trained model's average accuracy on the Places365 dataset is slightly lower by a few points), which indicates the differences in the comprehensiveness of different data sets. At the same time, the final results can be influenced by models even trained on the same dataset but in varying settings. For instance, the average accuracy of the pre-trained Densenet-121 from Pytorch is 0.43 higher than that from Timm Wightman (2019) on the ImageNet-to-Sketch benchmark.

**The computational complexity analysis of CPMS**   Our computational efficiency derives from more universal parent model searching. The efficiency of our parameter search method is primarily reflected in more effective information exchange across domains. Suppose we have $N$ tasks where each model can obtain the state that information from all other datasets is available to the model. This state can be referred to as the complete information exchange state. The faster we reach this state, the higher the efficiency of the information exchange. The parent model parameters in our method come from all the trained models, thus requiring at least two passes through all tasks, which amounts to $2N$ model training to achieve the state. In contrast, methods like MuNet and TAPS, which also require parameter search, derive the parent model parameters from only one trained model directly at a time. They need $N + (N-1) + ... + 1 = \frac{N(N+1)}{2}$ times training to reach the state. Therefore, we have reduced the complete information exchange from polynomial to constant complexity.

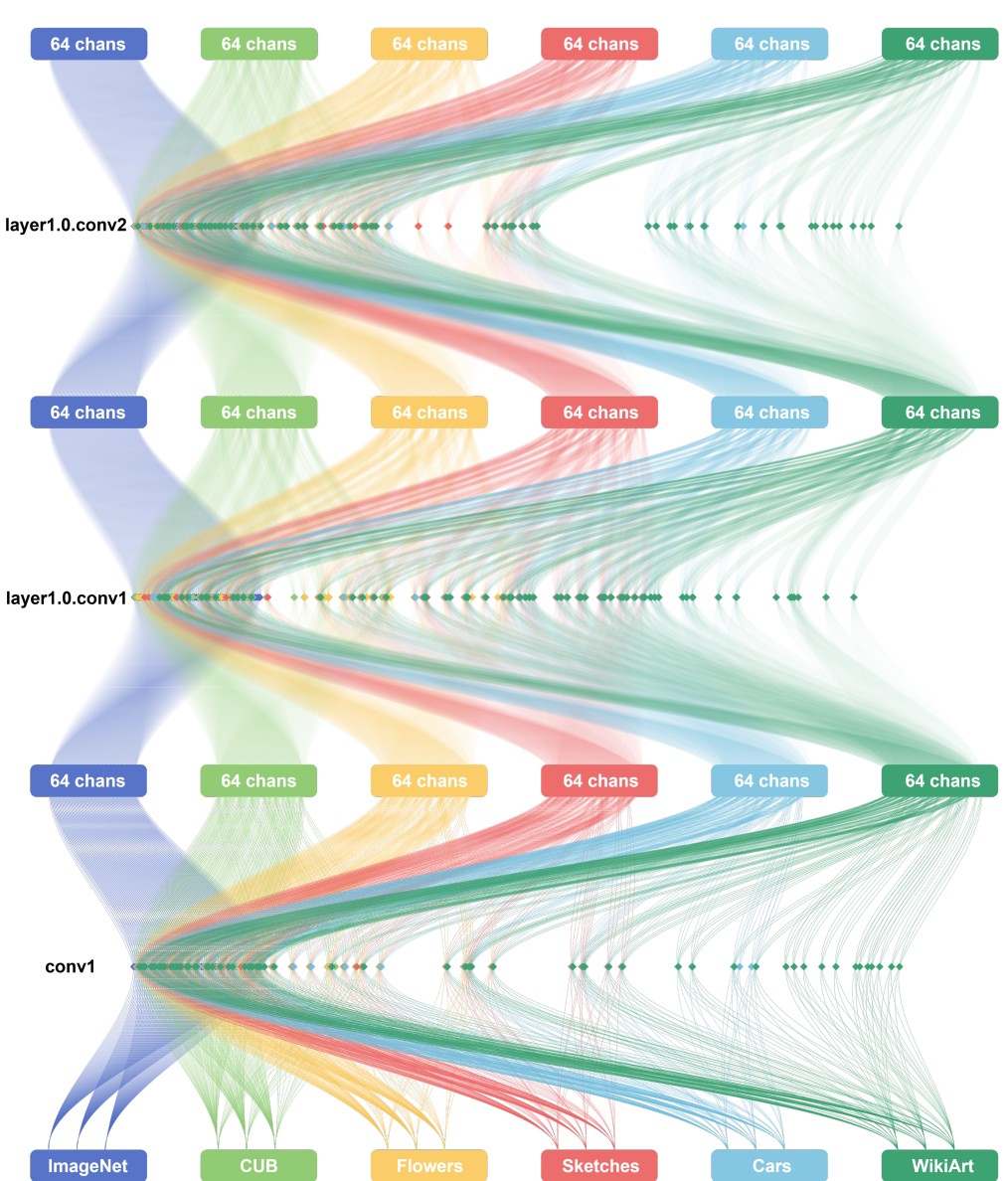

**Figure 8:** The structure of the first three layers of the supernet mentioned in Figure 6. Layers with more channels are difficult to visualize due to their complexity.

