# OpenReview forum: "CWPS: Efficient Channel-Wise Parameter Sharing for Knowledge Transfer"
_ICLR.cc/2025/Conference — ICLR 2025 Conference Withdrawn Submission_

### Official Review · Reviewer_jCg2 · 2024-11-03

**Soundness:** 2
**Presentation:** 2
**Contribution:** 2
**Rating:** 5
**Confidence:** 2

**Summary:**

This paper focuses on multi-task learning and multi-domain learning. Existing methods have not taken into account the characteristics of neurons, so a channel-wise parameter sharing method, CWPS, is proposed, achieving better results than the baseline.

**Strengths:**

1. CWPS has lower overhead for additional parameters compared to previous methods.
2. By using CPMS to determine the relationships between different tasks, it can effectively quantify these connections.

**Weaknesses:**

1. The paper argues that sharing at the neuron or channel level is important, but I think this requires more justification, as I didn’t see sufficient evidence for it in the paper. So, it doesn’t seem very different from other parameter-sharing methods, and the motivation behind the approach doesn’t seem very clear to me.
2. The overall design of the method appears relatively simple, and the comparison results lack novelty. More experiments are needed to uncover the core reasons behind the improvements this method brings.
3. The writing is poor.

**Questions:**

1. There is a lack of explanation regarding some baselines, as well as the approach to parameter sharing. Additionally, there are several prompt-like methods, such as DualPrompt.

---

> ### Author Response · Authors · 2024-11-25
> **Response to Reviewer jCg2**
>
> ### W1: Regarding Motivation
>
> Thank you for your comments. Our motivations are as follows:
>
> First, finer-grained parameter sharing signifies a more rational modeling approach. A common understanding is that neurons are the fundamental knowledge units of neural networks. Parameter sharing between models essentially represents the sharing of knowledge; from this perspective, choosing our method is reasonable.
>
> Second, finer-grained parameter sharing leads to more efficient memory consumption. Other methods, such as layer-based parameter sharing, isolate the entire layer's parameters if a layer is task-specific. This is not reasonable, as some channels within that layer may share similar patterns with other tasks but are not shared across tasks, leading to wasted parameters. In real-world deployments of multi-task networks, frequently reading different task models from memory(CPU) and disk is very time-consuming. Therefore, the optimal solution for multi-task networks is to store them directly in memory(GPU). Memory(GPU) is often one of the most valuable computational resources, and our multi-task network can save a significant amount of memory in this regard (as reflected by the Param Count metric).
>
> Finally, from a general perspective, neurons represent the most universal structure among various models. A common deep network may lack residual paths or attention mechanisms, but it will certainly have a structure based on neurons. Therefore, algorithms for parameter sharing at the neuron level are applicable across these structures.
>
> ### W2: Regarding expriments
>
> We have presented more results in APPENDIX.
>
> ### Q1: Regarding baseline
>
> Our baseline results are from [TAPS](https://openaccess.thecvf.com/content/CVPR2022/papers/Wallingford_Task_Adaptive_Parameter_Sharing_for_Multi-Task_Learning_CVPR_2022_paper.pdf). We will add detailed references in our paper.
>
> Additionally, since our method starts from any arbitrary pre-trained network to achieve efficient knowledge transfer for other tasks, it conflicts with DualPrompt, which is based on the transformer mechanism and forms task prompts for each class to enable continual learning. We will include a comparative reference to this in the paper:
>
> Methods based on prompts (Wang et al., 2022) attempt to establish a prompt for each task and category to measure their correlations, quantifying the relationships between tasks. However, due to the limitations of the transformer architecture, this approach still has certain application constraints on other backbones, such as convolutional networks.

---

> > ### Comment · Reviewer_jCg2 · 2024-11-27
> > **Thanks for the responses**
> >
> > Thanks for the responses, and I decide to keep my ratings.

---

### Official Review · Reviewer_bTwu · 2024-11-04

**Soundness:** 2
**Presentation:** 2
**Contribution:** 2
**Rating:** 5
**Confidence:** 4

**Summary:**

The paper proposed a method for knowledge sharing to tackle the problem of transfer learning. The method uses existing models as a knowledge pool, from which a layer-wise composition is constructed tailored to the new task to be solved. This composition is referred to as the parent model. Together with the parent model, a child model of the same architecture is trained. The two models are combined channel-wise using learnable masks. Experiments on multiple benchmarks are conducted to demonstrate the effectiveness of the approach.

**Strengths:**

1. The paper addresses an important problem in transfer learning. With more and more large foundation models available to the public, these models contribute a very large pool of knowledge learned through vast volume of data. The techniques investigated in this paper can be greatly beneficial to tapping into the potential of such models.

2. The proposed method seems fairly universal. Even though the experiments in the paper are mostly conducted on CNNs, the method can be potentially applied to state-of-the-art transformer networks as well.

**Weaknesses:**

1. The training pipeline seems unnecessarily convoluted and impractical. If I understand this correctly, given a new task, a child model first needs to be trained for a short period of time. This child model is then used as reference to construct the parent model. Afterwards, the two models are combined with learnable masks to tackle the aforementioned new task. Compared to standard fine-tuning, the advantage of the proposed method appears to be its parameter efficiency. However, the parameter count itself is not the most faithful reflection of the complexity of the model. Stats such as memory consumption and training time are more important, and accurately reflect the Flops needed to train the model. Given that the performance of the proposed method is generally worse than fine-tuning, the contribution of the work is less than convincing.

2. A major motivation of transfer learning is to exploit the knowledge learned on large volume of data and apply it to domain where data may be scarce. In this paper, however, it appears that all data of the target task is assumed to be available. This significantly reduces the practical value of the work. On the other hand, if the effectiveness of the proposed method can be demonstrated in a few-shot setting, where only a small number of labelled examples are available, I think this can make the paper much stronger. For a point of reference, I believe a recent work by Zhang et al. (2024) shows that a learnable linear combination of models in the weight space can achieve very strong performance in few-shot learning and also test-time adaptation. In the latter case, no labelled data is available. I would encourage the authors to compared against this method.

3. The writing of the paper is not the best, with numerous typos. For instance, in L201, I think the authors meant "existing" instead of "exiting".

Refs:
- Zhang et al., Knowledge Composition using Task Vectors with Learned Anisotropic Scaling. NeurIPS 2024

**Questions:**

N/A

---

> ### Author Response · Authors · 2024-11-25
> **Response to Reviewer bTwu**
>
> ### W1: Regarding Contributions
>
> We will address this issue from two aspects: memory usage and inference speed.
>
> First, our method is primarily aimed at multi-task networks in memory-constrained applications. In real-world deployments of multi-task networks, frequently reading different task models from memory(CPU) and disk is very time-consuming. Therefore, the optimal solution for multi-task networks is to store them directly in memory(GPU). Memory(GPU) is often one of the most valuable computational resources, and our multi-task network can save a significant amount of memory in this regard (as reflected by the Param Count metric), enhancing its deployment flexibility.
>
> Regarding inference speed, our method does not reduce the inference speed of the backbone. Existing multi-task methods based on [adapters](https://papers.nips.cc/paper_files/paper/2017/hash/e7b24b112a44fdd9ee93bdf998c6ca0e-Abstract.html), [pruning](https://openaccess.thecvf.com/content/CVPR2021/papers/Yan_DER_Dynamically_Expandable_Representation_for_Class_Incremental_Learning_CVPR_2021_paper.pdf), etc., significantly modify the model structure during design, which can render certain hardware acceleration methods for general backbones ineffective. In contrast, from the perspective of each task's model, our method remains structurally consistent with the backbone (the number of layers remains unchanged, and the number of channels per layer matches the backbone). This makes our approach more reasonable in practical applications.
>
> We will include these discussions in the revised version of the paper.
>
> ### W2: Regarding Data and More Comparisons
>
> Regarding data availability: In incremental learning scenarios, data is indeed limited for our method. Each time a new task is trained, data from previous tasks is unavailable to our algorithm. Additionally, due to the design of the ImageNet-to-Sketch benchmark, the amount of data for the tasks to be transferred is very small, necessitating an efficient knowledge transfer algorithm to utilize the knowledge of the pre-trained model. Our algorithm assumes that all data is available only when iterating in a multi-task scenario.
>
> More Comparisons: Thank you for providing a relevant work. We carefully read the paper "Task Vectors with Learned Anisotropic Scaling." Unfortunately, since the project is not fully open-sourced, we are unable to make a quantitative comparison. However, we have added relevant discussions in the paper:
>
> The method based on Task Vectors (Zhang et al., 2024) was very inspiring for our design approach. Our method essentially seeks to identify a finer-grained task vector that recognizes the underlying common information between tasks.
>
> ### W3: Regarding Writing
>
> Thank you for your careful review. We will address this issue in the revised version.

---

> > ### Comment · Reviewer_bTwu · 2024-11-26
> >
> > Thank you for the response.
> >
> > Having few learnable parameters is generally correlated but does not necessarily lead to low memory consumption. The proposed method combines a parent model and a child model of the same architecture, both of which need to be placed on the GPU. This will introduce additional memory consumption compared to training a single model. I suggest the authors use the profiling tools in packages such as PyTorch to measure the peak memory consumption in training. Similarly, the GFlop count precisely measures the computational complexity of the model, and is more objective than inference speed which can vary across different GPUs. These stats will provide a more holistic view of the model's performance, which I believe can make the paper stronger.
> >
> > Nevertheless, it appears that the proposed method still underperforms than standard fine-tuning. It's unclear what the advantage is for using a seemingly complicated method if it does not outperform a simple fine-tuning baseline. This is why I suggested maybe few-shot adaptation is a more suitable scenario. Because fine-tuning requires a large amount of data to achieve strong results and good generalisation, while transfer learning may still achieve good performance with scarce labelled data.

---

### Official Review · Reviewer_VFu2 · 2024-11-04

**Soundness:** 2
**Presentation:** 2
**Contribution:** 2
**Rating:** 3
**Confidence:** 4

**Summary:**

This paper introduces Channel-Wise Parameter Sharing (CWPS), a novel approach to knowledge transfer that enhances the efficiency and effectiveness of sharing parameters across different tasks or new data. Traditional methods like fine-tuning and layer-wise parameter sharing often provide coarse-grained solutions, which struggle to effectively identify shared parameters, thus limiting performance and efficiency. CWPS addresses these limitations by introducing fine-grained parameter sharing, achieved by refining the granularity from layers to neurons, allowing for the explicit composition of model neurons and utilization of knowledge from previous tasks. The paper also presents an effective search strategy to minimize computational costs and simplify the determination of shared weights. CWPS is designed to be comprehensive, plug-and-play, and has strong composability and generalization abilities, making it theoretically applicable to any network with linear and convolution layers. The method is evaluated across various datasets in incremental learning and multi-task learning scenarios, demonstrating superior precision-to-parameter ratio performance compared to existing methods, regardless of the backbone used.

**Strengths:**

1.Innovative Approach: CWPS offers a new perspective on knowledge transfer by enabling fine-grained parameter sharing, which is a significant departure from traditional coarse-grained methods.

2.Enhanced Efficiency: The method is designed to be efficient in terms of parameter sharing, which can potentially lead to better performance and faster training times.

3.Comprehensive and Plug-and-Play: CWPS is presented as a comprehensive solution that can be easily integrated into existing networks without the need for extensive modifications.

**Weaknesses:**

1.Since the supplementary materials of the paper include experiments with the Transformer model, it seems inconsistent that the main text focuses on convolution as the primary subject of discussion, rather than the Transformer. Moreover, the performance of Vision Transformer (ViT) in Table 5 of the appendix shows suboptimal results, which somewhat undermines the argument that the algorithm can generalize across different backbones effectively.

2.Why were experiments not conducted on datasets of a scale comparable to ImageNet in Tables 1 and 2? Observing the right side of Figure 4, it appears that all datasets have some relation to ImageNet. Therefore, wouldn't conducting a downstream task on a dataset of similar scale to ImageNet, but not necessarily entirely relevant, further demonstrate that the method's effectiveness is not solely due to knowledge transfer from ImageNet, but rather that the algorithmic design itself contributes significantly to the results?

**Questions:**

Would it be possible to visualize which channels are masked at each layer and whether there is a discernible pattern?

---

> ### Author Response · Authors · 2024-11-25
> **Response to Reviewer VFu2**
>
> ### W1: Regarding Vision Transformer Results
>
> Thank you for your comments. We have provided the experimental results based on ViT in Table 5 in the appendix. Although our baseline method TAPS is nearly consistent with our method in average accuracy, TAPS uses approximately 4x the number of parameters, while our method only requires 2.6x the number of parameters. We did not include the parameter count as a column in the table, so we will emphasize this point in revised versions.
>
> ### W2: Regarding Experiments on ImageNet
>
> Thank you for your comments. The goal of our method is to transfer a model trained on a large-scale dataset (ImageNet) to a smaller task using as few training resources and additional parameters as possible.
>
> First, a general fact is that in real-world applications, there often isn't sufficient large-scale data available to solve a small task, so pre-trained models are used to improve the performance of the model for that task through knowledge transfer.
>
> Additionally, unfortunately, the computational resources required for large-scale datasets exceed our capabilities and go against the original design intent of our method for task transfer.
>
> In summary, we did not conduct experiments on datasets of the same scale as ImageNet, and we appreciate your understanding.
>
> ### Q1: Regarding Visualization
>
> Thank you for your comments. In Figure 8 of the appendix, we further visualize the channel-sharing/task-specific situation for the first four layers after the execution of our method. Due to the large number of channels in each layer of the network, we did not fully visualize the subsequent layers.

---

### Official Review · Reviewer_KQbP · 2024-11-08

**Soundness:** 3
**Presentation:** 2
**Contribution:** 3
**Rating:** 6
**Confidence:** 3

**Summary:**

Current methods for knowledge transfer such as fine-tuning or weight sharing only offer coarse-grained sharing solutions and do not search for optimal parameters for sharing. This paper introduces Channel-Wise Parameter Sharing (CWPS) for efficient knowledge transfer. By refining the granularity of shared parameters from the layer level to the neuron level, they achieve fine-grained parameter sharing to address the coarse-grained problem. The authors also propose a simple method to search for suitable parameters for sharing. The proposed method achieves state-of-the-art results on several benchmarks.

**Strengths:**

- CWPS demonstrates high performance on various tasks and scenarios.
- The proposed framework is simple but effective.

**Weaknesses:**

- The experiment is mainly done using ResNet-50, lacking the experiment using other models such as DenseNet and EfficientNet,  or even transformer-based or MLP-based models.
- Lack of experiments with different depths of the model

**Questions:**

1. Can the authors explain more about parameter count?

---

> ### Author Response · Authors · 2024-11-25
> **Response to Reviewer KQbP**
>
> ### W1: Regarding Other Model Structures
> Thank you for your comments. We have provided experimental results for more types of networks in the appendix, such as the results based on ViT in Table 5. Although our baseline method TAPS is nearly consistent with our method in average accuracy, TAPS uses approximately 4x the number of parameters, while our method only requires 2.6x the number of parameters.
>
> ### W2: Regarding Model Depth
> Thank you for your comments. We conducted ablation experiments using networks of different depths, and below are the results of our experiments:
>
> |Model|Param Count|Flowers|WikiArt|Sketch|Cars|CUB|
> |:-:|:-:|:-:|:-:|:-:|:-:|:-:|
> |ResNet-18|3.3x|92.62|69.77|78.78|88.86|80.47|
> |ResNet-34|3.4×|94.13|76.20|80.33|90.89|83.06|
> |ResNet-50|3.2x|94.74|76.25|81.29|91.80|83.90|
>
>
> Our method demonstrates a stable parameter-performance ratio.
>
> ### Regarding Parameter Counting
> Thank you for your comments. The parameter counting method in our paper is based on TAPS and counts in terms of the parameters of the pre-trained model. Specifically, under this counting method, if the pre-trained model has a parameter count of 10M and the final multi-task network parameters amount to 30M, then the parameter count for this method would be 3x.

---

### Author Response · Authors · 2024-11-25
**Response To All Reviewers**

Thank you to all the reviewers for their careful and responsible feedback. In this paper, we propose a fine-grained knowledge transfer strategy that broadly facilitates efficient parameter sharing between tasks across various backbone structures. **Reviewer KQbP** believes our method demonstrates "high performance" and is "effective." **Reviewer VFu2**, while expressing concerns about our task setup, considers our method to be "innovative" and possess "efficiency." **Reviewer bTwu** describes our approach as "universal" and addressing an "important" issue. **Reviewer jCg2** notes that our method "effectively" quantifies the relationships between tasks. While some reviewers have questioned the motivations and effectiveness of our method, I believe our responses will address their concerns.

---

### Note · Authors · 2025-02-05

I have read and agree with the venue's withdrawal policy on behalf of myself and my co-authors.

---

### Meta-Review · Area_Chair_xfZh · 2024-12-20

**Metareview:**

The paper proposes to share parameters at the neuron (channel level) as opposed to other method such as the layer level. This is comparatively fine grained when compared to existing methods.The method  employs learnable masks and a search strategy to optimize shared parameters, Results are shown on existing multi-task learning benchmarks such as ImageNet-to-Sketch and DomainNet benchmark. The reviewers all appreciate the proposed framework and its universality of being applicable to any network. However, there is a consensus that the experiments do not fully convince the reader about the effectiveness of the method. Given the ratings, the paper unfortunately is not ready for ICLR.

**Additional Comments On Reviewer Discussion:**

Reviewer VFu2 did not respond to the author response. However, the other reviewers responded but the response did not result in a favorable rating for the paper. Reviewer bTwu clearly outlined his reason for retaining his score as opposed to Reviewer jCg2 who decided to keep his rating without a clear explanation.

---

### Decision · Program_Chairs · 2025-01-22

Reject